# Responses of Germination to Light and to Far-Red Radiation—Can they be Predicted from Diaspores Size?

**Luís Silva Dias** ***, Elsa Ganhão** and **Alexandra Soveral Dias**

Department of Biology, University of Évora, Ap. 94, 7000-554 Évora, Portugal; emng@uevora.pt (E.G.); alxandra@uevora.pt (A.S.D.)
*  Correspondence: lsdias@uevora.pt

**Abstract:** This paper presents an update of a dataset of seed volumes previously released online and combines it with published data of the photoblastic response of germination of fruits or seeds (light or dark conditions), and of the effects of enhanced far-red radiation on germination. Some evidence was found to support that germination in larger diaspores might be indifferent to light or dark conditions. Similarly, germination in smaller diaspores might be inhibited by far-red radiation. However, the length, width, thickness, volume, shape, type of diaspore, or relative amplitude of volume is essentially useless to predict photoblastic responses or the effects of far-red radiation on germination of diaspores.

**Dataset:** Available as supplementary file and at http://home.uevora.pt/~lsdias/SVDS_03plus.csv.

**Dataset License:** CC-BY

**Keywords:** diaspore shape; diaspore size; inhibition of germination by far-red radiation; germination in light and dark

---

## 1. Introduction

Recently [1], a dataset of diaspore volumes was released and described, with volume calculated from length, width, and when available, thickness, abstracted from a variety of sources. In addition, the adequacy of minimum diaspore volume as a surrogate for size in studies of functional and ecological correlation of diaspores size was investigated using data published by Milberg et al. [2] (hereafter referred to as Milberg et al.) on the relationship between light requirement for germination and diaspores size. The original relationship, in which diaspores size was expressed by average mass, was then compared with a recalculated relationship with size expressed by the minimum volume, and the latter was found to be essentially equivalent to the former.

However, results of Milberg et al. heavily relied on positively photoblastic diaspores, the germination of 85% of them being inhibited by dark; only light or dark conditions were tested, and diaspores size expressed by average mass could account only for less than one third of the variation of the photoblastic response of diaspores germination.

In several ways, the present work constitutes a follow up to their investigation on size correlation of germination responses to light. It combines data available in the dataset mentioned above with data published by Górski et al. [3–5] (hereafter referred to as Górski et al.) which still is, to our knowledge, the most extensive survey of photoblastic responses and of effects of far-red radiation on diaspore germination [6].

Thus, the objective of this work is to answer the two questions that constitute the title of this paper: can we use knowledge of diaspores size to predict their germination under light or dark conditions and under far-red enhanced radiation?

The period between embryo protrusion and the establishment of a fully independent plantlet, the seedling stage, is one of the most critical and vulnerable phases in the life of plants, light availability playing an undisputed role in it [7]. Nevertheless, it is beyond our purpose to investigate the role of light or dark and of far-red radiation in the events following diaspore germination, namely seedling early growth.

As discussed elsewhere [8], diaspore weight, despite its frequent use, is not a measure of diaspore size, only of diaspore mass. Conversely, diaspore volume is a real measure of diaspore size and minimum volume has been shown to be an adequate and reliable single-value representative of the size of diaspores [1]. Therefore, throughout this paper we will use minimum volume to represent size.

Changes in size and shape of diaspores, even if very small, can affect almost every aspect of diaspores dormancy and persistence in soil or of germination and subsequent growth and development of plants [9].

Effects of environmental factors on germination and the persistence of diaspores in soil are intimately connected aspects of plants functional ecology. A number of investigations, sometimes contradictory, have been reported on the role of size and of shape on diaspores persistence in soil in a variety of regions and environments [10–17], making the consideration of shape highly advisable, in addition to size, when investigating the response of diaspores to light and far-red radiation.

However, defining shape can be a contentious matter, even more so if shape quantification is intended, and a discussion of this issue is beyond the scope of this paper. Examples can be found in [8,18–20]. Throughout this paper we adopted the approach originally presented by Thompson et al. [10] while referring to it as departure from sphericity instead of shape.

Evidence exists that being a fruit or a seed may play a role in the persistence of diaspores in soil [10,11,13] and it was also shown that the relationship between the volume and the departure from sphericity of diaspores depends on them being fruits or seeds [8], prompting us to include this trait in the search for a relationship between diaspores' size and their germination in light or under far-red radiation.

Therefore, the question stated above can be rephrased in a more detailed way as: can we use knowledge of diaspore size, departure from sphericity, and type (fruit or seed) to predict germination under light or dark conditions or under far-red enhanced radiation?

## 2. Results

### 2.1. Photoblastic Responses

In positively photoblastic diaspores the minimum volume ranged from $1.286 \times 10^{-3}$ (*Phyteuma orbiculare*) to 5.079 mm$^3$ (*Scabiosa ochroleuca*) with a median of 0.134 mm$^3$. In indifferent diaspores it ranged from $4.719 \times 10^{-4}$ (*Spergula arvensis*) to 66.626 mm$^3$ (*Rheum rhaponticum*) with a median of 0.705 mm$^3$. In negatively photoblastic diaspores it ranged from 0.067 (*Poa palustris*) to 15.022 mm$^3$ (*Vicia faba*) with a median of 1.097 mm$^3$. Distribution of diaspores size (expressed as equivalent diameter) for each class of photoblastic response is presented separately for fruits and seeds in Figure 1a.

In positively photoblastic diaspores the departure from sphericity ranged from 0.023 (*Veronica gentianoides*) to 0.205 (*Phyteuma orbiculare*) with a median of 0.093. In indifferent diaspores it ranged from 0.020 (*Ipomoea hederacea*) to 0.190 (*Leontodon autumnalis*) with a median of 0.083. In negatively photoblastic diaspores it ranged from 0.028 (*Eschscholzia californica*) to 0.213 (*Tragopogon major*) with a median of 0.076. Distribution of departure from sphericity of diaspores for each class of photoblastic response is presented separately for fruits and seeds in Figure 1b.

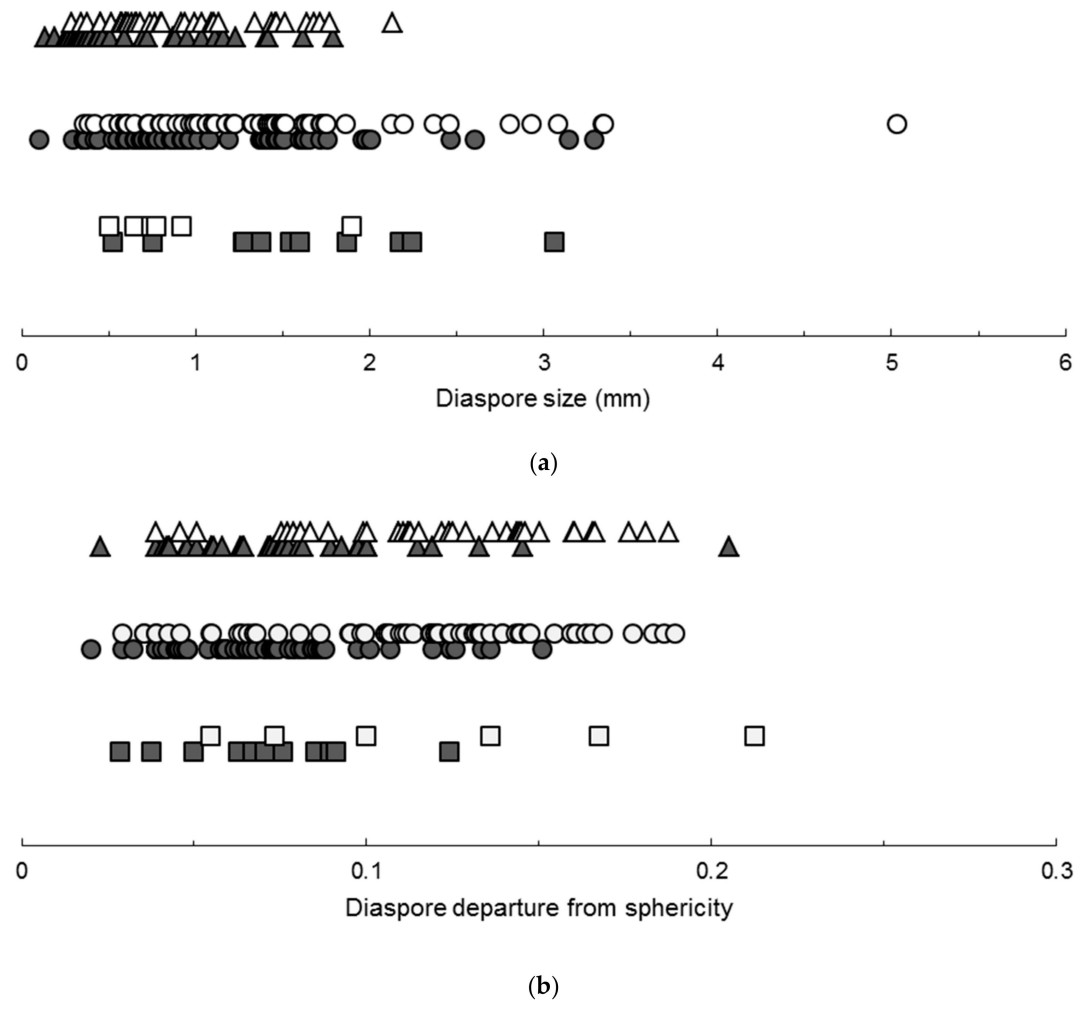

**Figure 1.** Distribution of positively (triangles), indifferent (circles), and negatively (squares) photoblastic diaspores separately for fruits (open symbols) and seeds (closed symbols) in relation to: (**a**) size (expressed as equivalent diameter); (**b**) departure from sphericity (expressed as population variance of linear dimensions normed so that length is unit).

Optimal decision trees were obtained in 24 out of the 100 independent runs of classification, corresponding for all practical purposes to three different decision trees. Two trees were obtained five times and one tree was obtained fourteen times.

All binary decision trees had only one variable. Pooling together the learning and the validation group, the most frequent tree had a mean value of misidentification of the photoblastic response of diaspores of 56%. It misidentified 53% of positively photoblastic diaspores (between 48% and 54% in the learning group, between 52% and 64% in the validation group); 14%–15% of indifferent diaspores (between 14% and 19% in the learning group, between 7% and 12% in the validation group); and it misidentified all negatively photoblastic diaspores. The same happened in the other decision trees, with negatively photoblastic diaspores usually identified as indifferent. The most frequent decision tree had only one step and can be expressed as a dichotomous key (in three trees the cutting value was 0.291 mm):

| | |
|---|---|
| 1. Diaspore thickness ≤ 0.292 mm | Positive photoblasty |
| Diaspore thickness > 0.292 mm | Indifferent photoblasty |

The best decision tree was only slightly better than the most frequent we have just described. Its mean value of misidentification of diaspores was 54% and had only one step involving the width (cutting value of 0.625 mm in four cases, of 0.674 mm in one case). Finally, the worst decision tree had

a mean value of misidentification of 60% and only one step involving the volume (cutting value of 0.022 mm$^3$ in two cases, of 0.023 mm$^3$ in three cases).

## 2.2. Effects of Far-Red Radiation

No stimulation of diaspores germination by far-red radiation was found. In diaspores indifferent to far-red radiation the minimum volume ranged from 0.027 (*Digitalis purpurea*) to 19.538 mm$^3$ (*Zea mays*) with a median of 1.437 mm$^3$. In diaspores inhibited by far-red radiation it ranged from $4.719 \times 10^{-4}$ (*Spergula arvensis*) to 66.626 mm$^3$ (*Rheum rhaponticum*) with a median of 0.275 mm$^3$. Distribution of diaspores size (expressed as equivalent diameter) for each class of response to far-red radiation is presented separately for fruits and seeds in Figure 2a. Location of diaspores with *RFR* = 1 (Equation (1)), thus with germination under canopy completely inhibited, is also shown.

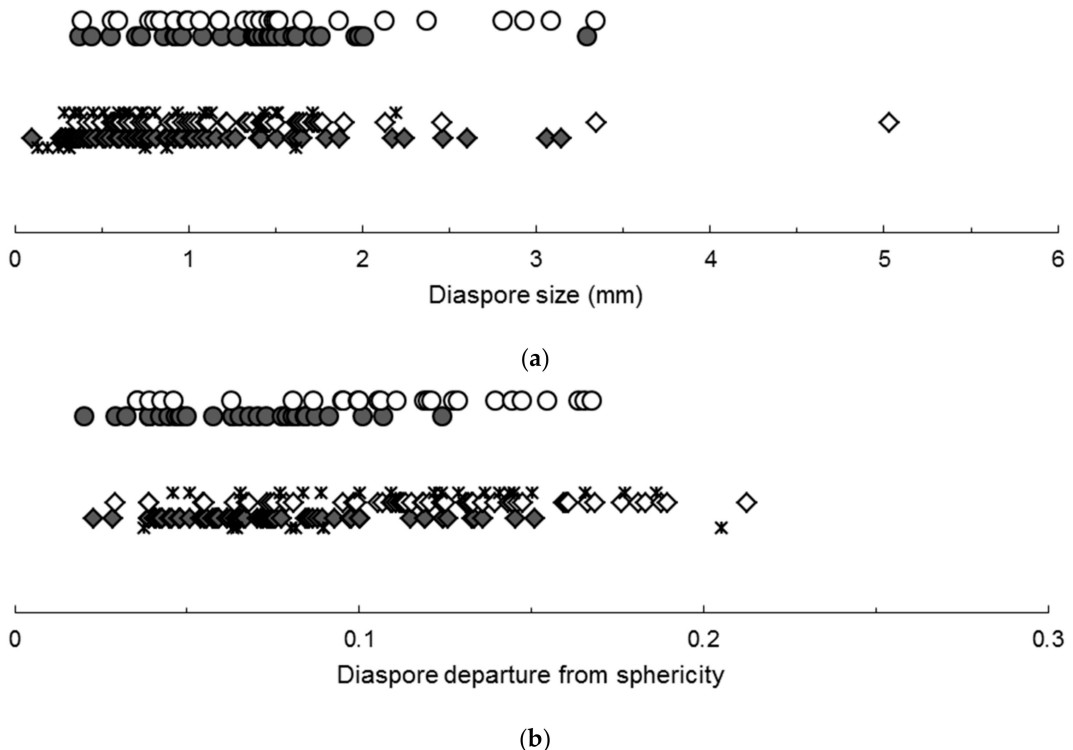

(**a**)

(**b**)

**Figure 2.** Distribution of diaspores with germination indifferent to (circles), and inhibited (diamonds) by far-red radiation, separately for fruits (open symbols) and seeds (closed symbols), in relation to: (**a**) size (expressed as equivalent diameter); (**b**) departure from sphericity (expressed as population variance of linear dimensions normed so that length is unit). Diaspores with germination completely inhibited by far-red radiation are represented by asterisks.

In diaspores with germination indifferent to far-red radiation the departure from sphericity ranged from 0.020 (*Ipomoea hederacea*) to 0.168 (*Solidago virga aurea*) with a median of 0.081. In diaspores inhibited by far-red radiation it ranged from 0.023 (*Veronica gentianoides*) to 0.213 (*Tragopogon major*) with a median of 0.089. Distribution of departure from sphericity of diaspores for each class of response to far-red radiation is presented separately for fruits and seeds in Figure 2b. Location of diaspores with *RFR* = 1 (Equation (1)), thus with germination under canopy completely inhibited, is also shown.

Only one optimal decision tree was obtained in the 100 independent runs of classification performed, with a pooled value of misidentification of 28% and frequently misidentifying diaspores with germination indifferent to far-red radiation as being inhibited by it.

## 3. Discussion

We are fully aware, as emphasized elsewhere [1], that the Seed Volume Dataset probably can never be fully finalized because the number of possible records of linear dimensions of diaspores is not finite. Therefore, using the Seed Volume Dataset as a source of diaspores' size necessarily implies that at any given moment diaspores' volumes are limited to those abstracted so far. However, at this stage of development of the Seed Volume Dataset we feel confident enough on the stability of the results and conclusions presented here based upon available data of diaspores size, not least because of the negligible impact of the present update of the Seed Volume Dataset on the relationship between diaspore volume and photoblastic data by Milberg et al. (Appendix A).

### 3.1. Photoblastic Responses

In the range of minimum volumes investigated in this work ($4.719 \times 10^{-4}$ mm$^3$ to 66.626 mm$^3$, four orders of magnitude, less than half of the number of orders of magnitude registered so far in the Seed Volume Dataset) germination of larger diaspores ($\geq$ 6.939 mm$^3$; 13 species) was always indifferent to the presence or absence of light, with the exception of *Vicia faba*, a negatively photoblastic species. Conversely, the germination of smaller diaspores ($\leq$ 0.022 mm$^3$; 18 species) was always inhibited in the dark (positive photoblasty), with the exception of *Spergula arvensis* which has the smallest diaspore of all, and *Lobelia erinus*, both indifferent to the presence or absence of light.

Therefore, some support can be found to the assertion that large diaspores are less dependent on light for germination than small diaspores [2]. Does this imply that a relationship exists between the photoblastic response of germination and size? As a consequence, can diaspores volume be used to predict their light requirements to germinate?

At the very least, the answer is hardly, if not utterly negative. In fact, diaspores with volume between 0.024 and 5.893 mm$^3$ (*Dianthus sinensis* and *Solanum melongena*, respectively) can be either positively photoblastic (91% of positively photoblastic species), indifferent (91% of indifferent species) or negatively photoblastic (94% of negatively photoblastic species), meaning that light, dark or indifference to light or to dark occur across the whole range of their size, thus making useless any attempt to predict photoblasty from size in the vast majority of species investigated.

The same occurs with even more intensity when the departure of sphericity of diaspores is examined, with an almost perfect coincidence in the range of departure from sphericity of positively, indifferent, and negatively photoblastic diaspores.

Finally, the type of diaspore also can hardly help the prediction of photoblastic responses. Neither do the simultaneous consideration of length, width, thickness, volume, departure from sphericity, diaspore type, and relative amplitude of volume.

Optimal decision trees were infrequent, varied widely with pseudo-random assignment of diaspores to the learning and validation groups, and had unacceptably large percentages of misidentification either in the best (54%) or in the most frequent decision tree (56%).

Until now we have viewed the inhibition of germination by dark (positive photoblasty) or by light (negative photoblasty) disregarding its intensity. However, functional and ecological implications of total or of partial inhibition of germination by light or by dark are not necessarily the same, but data for the level of inhibition were not available in Górski et al. and we cannot proceed further. By the contrary, data from Milberg et al. in which diaspores size was expressed by average mass shows that no relationship exists between complete inhibition of germination by dark (total positive photoblasty) and size.

It also provides some evidence (Appendix B) that inhibition of germination by light or by dark per se might not represent an important barrier to the recruitment of new individuals provided that inhibition is not complete and that the combination of diaspores output, diaspores germination and plantlet mortality are such that for each mother plant at least one plant survives and produces diaspores. It can even be hypothesized that positively and negatively photoblastic species might not differ that

much from indifferent species in their ability to explore different light conditions and environments, provided that inhibition of germination by dark or by light is not complete.

A completely different situation would result from a complete inhibition of germination in light or dark. However, one can ask whether it is reasonable to expect a complete light-mediated inhibition of germination in the field, whichever the light conditions, or whether such an inhibition is not, in large measure, an experimental artifact as suggested by the results of *Verbascum thapsus* described in Appendix B.

### 3.2. Effects of Far-Red Radiation

In general, and with the necessary adaptations, the discussion of photoblastic responses in the previous section can be used in the framework of the effects of far-red radiation on diaspores germination.

Germination of smaller diaspores (<0.027 mm$^3$; 23 species) was always inhibited by enhanced far-red radiation, but no evidence exists as to the effects on the germination of larger seeds. In fact, 86% of species inhibited by far-red radiation and all indifferent species have diaspores with volume ≥0.027 mm$^3$.

Support for a lesser dependency of germination of large diaspores from full light conditions are even thinner than when light and dark conditions were investigated and attempts to predict germination responses to red or far-red radiation using diaspores size are equally or even more useless.

The same occurs when the shape of diaspores is examined, with an almost perfect coincidence in the range of departure from sphericity of the majority of diaspores indifferent and inhibited by far-red radiation. Despite that the germination of diaspores with departure from sphericity larger than 0.168 (10 species) was always inhibited by far-red radiation, 94% of species inhibited by far-red radiation and all indifferent species had departures from sphericity between 0.020 and 0.168.

Finally, the type of diaspore also can hardly help the prediction of responses to red or far-red radiation. Neither do the simultaneous consideration of length, width, thickness, volume, departure from sphericity, diaspore type, and relative amplitude of volume. Only one optimal decision was obtained, meaning that it resulted from a rare combination of pseudo-random selection of diaspores for the learning and validation groups.

Viewing separately the 27 species in which germination is completely inhibited by far-red radiation, again no evidence was found of a predictive relationship between size, departure from sphericity or diaspores type and germination.

However, in the remaining species with germination incompletely inhibited by far-red radiation, and assuming 90% diaspores loss before germination (by dormancy, burying, decay, predation, or any other cause) and, whatever the cause, 90% of plants arising from germinated diaspores dying before a mature plant producing diaspore is formed (probably gross over-estimates of loss and death), in some cases these assumptions might be too high, thus risking the recruitment of a new generation. It is the case of *Plantago media*, where annual diaspores production per plant can be 830 [21], which combined with 1% germination would result in less than 1 new plant per mother plant. Conversely, no such risks seem to exist in other species with germination inhibited by far-red radiation for which we found data of seed output [21]. For example, *Agrostemma githago* (300 seeds, 61% germination, 1–2 new plants per mother plant) or *Hypericum perforatum* (26,000 seeds, 7% germination, 18 new plants per mother plant).

Again, inhibition of germination by radiation conditions, if not complete, might not represent a barrier to the recruitment of new individuals provided that the combination of diaspores output, diaspores germination, and plantlet mortality are such that for each mother plant at least one plant survives and produces diaspores.

## 4. Materials and Methods

### *4.1. Sources of Data*

Data used in this work derive from two different sets of data. One is a dataset of diaspores volume (seeds or fruits) calculated from length, width, and when available, thickness, which is released here and constitutes the fourth update of Seed Volume Dataset. The other is a set of data of germination of 240 species published between 1975 and 1978 by Górski et al.

#### 4.1.1. Seed Volume Dataset

The Seed Volume Dataset was organized in tabular form in an MS Excel 2010® spreadsheet, which is available in the Supplementary Materials (Supplementary S1) as a non-proprietary comma-separated values (CSV) format file. The version presented here is also available in the same format at [22] and updates the version presented in [1]. Future updates will also be available at that link. However, one addition was made to the previous version, and a new column was introduced in which all corrections made to the third update are noticed and briefly explained. A full and detailed description of the Seed Volume Dataset, including the rationale for using the minimum volume as surrogate of size, can be found in [1]. The rationale and methods for the determination of volume from linear dimensions can be found in [23,24]. The rationale and methods for the determination of departure from sphericity as a surrogate of shape can be found in [8,10,25]. For a better visualization, in figures we will represent diaspore volume by the diameter that a sphere with the same volume would have (equivalent diameter).

Briefly, in the fourth update, released now, the number of entries increased from 24,208 to 29,412, with valid nomenclatural entries representing 62% of entries and synonyms 38%. The number of species increased from 6544 to 7402, belonging to 222 families and 55 orders. All gaps in the numbering of entries and of sources of entries were filled and disappeared.

#### 4.1.2. Germination Dataset

In a series of articles published between 1975 and 1978, Górski et al. reported the results of three series of experiments, performed in the summers of 1974 [3], 1975 [4], and in the summer and early autumn of 1976 [5], of diaspores germination responses to light.

In the first experiment diaspores were sown in Petri dishes and tested for their sensitivity to white light either by keeping them inside light-tight boxes, or by exposing them to diffuse natural light in a room near a window [3] or on the edge of a window facing north [4]. Room temperature was controlled or partly controlled and maintained at 18 ± 1.5 or 19 ± 2 °C [3,4]. Differences of germination between treatments allowed the definition of diaspores as positively photoblastic, indifferent, or negatively photoblastic, but no quantitative germination data of the light/dark experiment was provided.

In the second experiment, diaspores were sown in Petri dishes on two layers of white flannel and one or two Whatman filters and moistened with distilled water. Distilled water was added during the experiments as needed. Dishes were placed either under rhubarb, rye, barley, dense currant and wild shrub canopies [3], or under dense rhubarb canopy only [4,5]. Control dishes were placed in wooden framework boxes permitting no sunflecks. Mean daily temperature ranged from 12 to 24 °C. Global radiation (300–3000 nm) inside boxes was controlled and kept approximately at the level measured under canopies. Under canopies ratios of short-wave (red, below 700 nm) to long-wave (far-red, above 700 nm) radiation varied from 0.10 to 0.14 while in boxes the same ratio was about 1.25.

Depending upon the species, final germination was recorded between 4 and 45 days after sowing. With the possible exception of the experiment performed in 1974, no treatments promoting germination were done and results were only given for species with at least 25% germination in the control. Inhibition of germination by far-red radiation was determined by Student's *t* tests at a significant level $P = 0.05$.

Overall, 240 species from 31 families (11 families with one species, 2 families with two species, 18 families with three or more species) were tested in the three series of experiments. Asteracea, Poaceae, Caryophyllacea, and Brassicaceae were the families with more species, together accounting for 55% of them. Positive photoblasty was found in 36% of species, indifferent photoblasty in 57%, and negative photoblasty in 7%. Five families of the 18 with three or more species were represented in the three types of photoblasty (Asteraceae, Poaceae, Ranunculaceae, Scrophulariaceae, and Solanaceae). Inhibition of germination by far-red radiation was found in 74% of species, indifference in 26%. Fourteen families of the 20 with two or more species were represented in the two types of response to far-red radiation (Amaranthaceae, Asteraceae, Brassicaceae, Campanulaceae, Caryophyllaceae, Chenopodiaceae, Fabaceae, Lamiaceae, Liliaceae, Plantaginaceae, Poaceae, Polygonaceae, Scrophulariaceae, and Solanaceae).

### 4.1.3. Combining Seed Volume and Germination Datasets

Despite the large number of species in the Seed Size Dataset (around 7,400, not counting synonyms) and the systematic check for synonyms in the Seed Size Dataset and in World Flora Online [26] we could find volumes for no more than 222 of the 240 species tested by Górski et al. Two or more entries were always available except in one species (*Campanula lactiflora*). Asteracea, Poaceae, Caryophyllacea, and Brassicaceae were again the families with more species accounting for 56% of them. The full list of the 222 species with data of photoblasty and germination in controls or under canopies plus additional data of linear dimensions, volume, departure from sphericity, relative amplitude of volume, and classification of diaspore as fruit or seed are available in the Supplementary Materials (Table S1).

With the reduction of the number of species available, Gesneraceae (with one species) disappeared and the number of families reduced to 30 (10 families with one species, 2 families with two species, 18 families with three or more species). Positive photoblasty was found in 35% of species, indifferent photoblasty in 58%, and negative photoblasty in 8%, closely resembling the original distribution in Górski et al. The same five families with three or more species were represented in the three types of photoblasty. Inhibition of germination by far-red radiation was found in 73% of species, indifference in 27%, again closely resembling the original distribution. Thirteen families out of 20 with two or more species were represented in the two types of response to far-red radiation (only Solanaceae disappeared).

Górski et al. expressed the effects of far-red radiation by the ratio between germination under plant canopy ($Lp$) and in the diffuse light control ($Ld$). However, this type of ratio is not symmetric in relation to $Ld = Lp$. Therefore, we adopted an alternative and symmetric ratio, similar to relative light germination used in [2], which can be expressed as:

$$RFR = Ld/( Ld + Lp ), \tag{1}$$

where *RFR* stands for relative effects of far-red radiation, $Ld$ and $Lp$ as above. *RFR* varies between $RFR = 0$ ($Ld = 0$, diaspores germinate only under plant canopies) and $RFR = 1$ ($Lp = 0$, diaspores germinate only in the diffuse light control, meaning complete inhibition of germination by far-red radiation).

Classification of diaspores as positively, indifferent, or negatively photoblastic followed Górski et al. criteria. Classification of indifference or inhibition of germination followed the results of Student´s *t* tests in Górski et al. Non-significant differences between $Ld$ and $Lp$ were taken to represent indifference to far-red radiation, significant differences to represent inhibition by far-red radiation (whenever significant differences occurred, $Ld$ exceed $Lp$).

### 4.2. Data Analyses

A good part of data analysis involved plotting the distribution of the three types of photoblastic responses of germination or of the two types of responses of germination to red or far-red radiation against diaspore size expressed by volume or diaspore shape expressed as departure from sphericity. However, a clear representation of putative correlation between several morphological traits of

diaspores acting simultaneously or separately and the responses of germination to light or to far-red radiation is beyond the ability of such plots.

Therefore, we used a non-parametric tree-structured classification method [27–29] to investigate whether and how a set of decision rules could be established to predict photoblastic responses and far-red radiation effects on diaspores germination. This method is a powerful and flexible classification tool using recursive and iterative procedures and is preferred to discriminant analysis or to logistic regression because it can be applied to any data structure handling ordered and categorical variables in a simple and natural way; it makes powerful use of conditional information when nonhomogeneous relationships have to be handled, does automatic stepwise variable selection and complexity reduction, is invariant under all monotone transformations of individual ordered variables, is extremely robust with respect to outliers and to misclassifications of input data, accommodates interactions without prior selection of variables, provides not only a classification but also estimates of misclassification probabilities, and gives easily understood and interpreted information regarding the predictive structure of data. It also allows for different costs of misclassification and results in rules of decision that are easy to understand and apply [27,29].

Decision trees to identify photoblastic responses or effects of far-red radiation were generated using SPAD Data Mining and Text Mining, v. 6.5.0 (SPAD, Paris, France). Seven candidate explanatory variables of the three qualitative photoblastic responses (positively, indifferent, and negatively photoblastic) or of the two qualitative responses to far-red radiation (indifference and inhibition) were used: diaspore length, width, thickness, volume, departure from sphericity expressed as variance, and diaspore type (fruit or seed), plus relative amplitude of volume (*RAV*) adopted as a surrogate of diaspore plasticity and expressed as:

$$RAV = (\ VOL_{max} - VOL_{min}\ )/VOL_{min}, \tag{2}$$

where $VOL_{max}$ and $VOL_{min}$ stand for the maximum and minimum volume respectively. Relative costs of misclassification were kept constant and unitary; 100 independent runs generated pseudo-randomly were done, always assigning proportionally 33% of records to the validation group; sub-optimal decision trees were always discarded.

The relationship between relative light germination and diaspore size in Appendix A was investigated by stepwise least squares regression without replication and an experiment-wise error rate for coefficients of 0.05 calculated by the Dunn–Šidák method [30]. Polynomials were used as candidate models and included up to the third power of the explanatory variable either untransformed or logarithmically transformed (base 10). Regression analyses were done with Statgraphics 4.2 (STSC, Inc., Rockville, MD, USA).

## 5. Conclusions

Some evidence was found on a greater requirement for light to germinate in smaller diaspores, simultaneously with a lesser requirement for light in larger diaspores, which does not translate into an ability of diaspore size to predict their photoblastic response. In fact, the large majority of diaspores investigated can be positively, indifferent, or negatively photoblastic regardless of size. Incidentally, the smaller diaspore of all (almost three times smaller than the next in size) had an indifferent response to the presence or absence of light. Diaspore shape and diaspore type were no better as predictors of photoblastic responses of germination. The same applies to the other morphologic traits investigated simultaneously for their predictive power.

A slightly different pattern occurs when the effects of far-red radiation are investigated, despite that the general picture is essentially the same. Some evidences of inhibition of germination by far-red radiation in smaller diaspores and of diaspores with shape departing more from sphericity, but not of diaspore type, were found. Nevertheless, the same reasoning used in relation to the response of germination to light/dark conditions applies here even more intensely. The large majority of diaspores investigated can be indifferent or have the germination inhibited by far-red radiation regardless of

their size and shape, and therefore these traits were useless to predict germination responses to far-red radiation. The same applies to the other morphologic traits investigated simultaneously for their predictive power.

**Supplementary Materials:** The following are available online at http://www.mdpi.com/2306-5729/5/2/49/s1, Supplementary S1 Seed Volume Dataset update 04 March 2020; Table S1: Species (synonymy used inside brackets), families, type of photoblasty, percentage of germination of diaspores in diffuse white light (Ld) or under plant canopy (Lp), result of significance tests (SIG) for the difference between Ld and Lp (* significant, and *ns* not significant at $P = 0.05$), and source of germination data. Length (L), width (W), thickness (T), volume (VOL), departure from sphericity (VAR), relative amplitude of volume (AMP), and diaspore type.

**Author Contributions:** Conceptualization, data curation, formal analyses, methodology, resources, L.S.D.; Investigation, E.G.; Writing—original draft preparation, writing—review and editing, A.S.D. and L.S.D. All authors have read and agreed to the published version of the manuscript.

**Funding:** This research received no external funding.

**Acknowledgments:** We thank the Herbarium of University of Évora for administrative and technical support of the Seed Volume Dataset.

**Conflicts of Interest:** The authors declare no conflict of interest.

## Appendix A

In relation to the previous release of the Seed Volume Dataset [1], this fourth update has new entries of linear dimensions of diaspores for two species of Milberg et al. previously absent (*Chenopodium polyspermum* and *C. suecicum*) and new entries for 23 species already present. Among the latter, in six species (*Centaurea cyanus*, *Conyza canadensis*, *Lactuca serriola*, *Lamium amplexicaule*, *Lapsana communis*, and *Myosotis arvensis*) the volume of diaspore was changed because a smaller value was found. Therefore, we recalculated the relationship between minimum volume of diaspores and the relative light germination expressed by Milberg et al. as:

$$RLG = Gl/(Gd + Gl), \tag{A1}$$

where *Gl* and *Gd* are the germination percentage in light and in darkness respectively.

The fitted equation was:

$$RLG = 0.7123 - 0.0777 \log_{10} VOL_{\min}, \tag{A2}$$

where $VOL_{\min}$ is the minimum diaspore volume in mm$^3$. The equation was significant ($P = 0.0235$), with all coefficients having $P \le 0.0235$. The coefficient of determination was 0.1044 (0.0854 when adjusted for degrees of freedom). Overall, and despite the addition of two new species and new data for 23 species (49% of species) involving the replacement of the minimum volume in six species (13% of species), the recalculated equation presents only negligible differences in comparison to the equation fitted formerly in [1] in which the constant was 0.0706 and the first degree coefficient was −0.0793.

## Appendix B

Adopting the criteria presented in Milberg et al., diaspores with $RLG > 0.55$ qualify as positively photoblastic, with $0.45 < RLG < 0.55$ as indifferent, and with $RLG \le 0.40$ as negatively photoblastic, *RLG* standing for relative light germination (Equation (A1)). Therefore, we reexamined the data of Milberg et al. using their original weight values and grouping diaspores according to their photoblastic response (Figure A1). The location of diaspores with $RLG = 1$ (Equation (A1)), thus with germination in dark completely inhibited, is also shown.

Positive photoblasty was found in 85% of species, indifference in 11%, and negative photoblasty in only 4%. No species showed complete negative photoblasty, but seven species showed complete positive photoblasty (*Chamomilla recutita*, *Polygonum persicaria*, *Rumex crispus*, *R. longifolius*, *R. obtusifolius*, *Verbascum thapsus*, and *Veronica arvensis*). However, no clear relationship between diaspores' mass and complete inhibition of germination in dark is apparent.

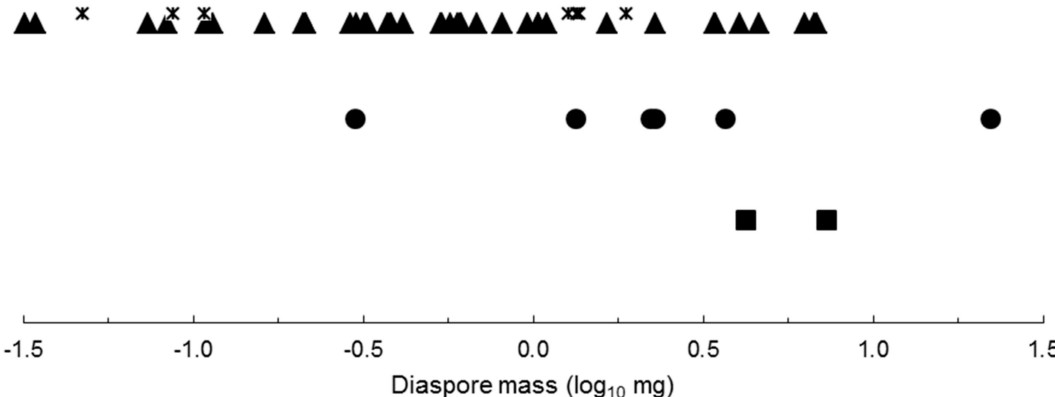

**Figure A1.** Mass distribution of positively (triangles), indifferent (circles), and negatively (squares) photoblastic diaspores. Diaspores with germination completely inhibited in dark are represented by asterisks.

According to Milberg et al., *Conyza canadensis* is a positively photoblastic species, with *RLG* = 0.97, resulting from 25% of germination in light and 0.773% in dark, which is undoubtedly a very low germination rate. However, a single plant of *C. canadensis* can produce annually some 250,000 diaspores [21]. Assuming that all diaspores are in dark conditions, that as much as 90% of them are lost before germination (by dormancy, burying, decay, predation, or any other cause) and that, whatever the cause, as much as 90% of plants arising from germinated diaspores die before a mature plant producing diaspore is formed, which are probably two gross over-estimates of loss and mortality, the new generation would amount to about 1.9 new plants per mother plant, despite the very low percentage of germination in dark of *C. canadensis*. Therefore, in positively photoblastic species, germination in the dark may allow the recruitment of enough new individuals to avoid local extinctions.

According to Milberg et al., *Verbascum thapsus* is a positively photoblastic species, with *RLG* = 1, which necessarily means that germination in the dark *Gd* = 0%. Nevertheless, germination in the dark for this species has been reported to be 0%–1% depending upon the time after harvest and conditions during the incubation of diaspores [31]. Combining the non-null percentage of germination with the diaspore output (104,000 diaspores) known for this species [21] and the probabilities of loss and dead presented above, the new generation would amount to about 10 new plants per mother plant. However, temperature, not light, is credited as being the most important environmental factor after water availability in diaspores' germination [32]. Germination of *V. thapsus* in dark at or below 19 °C was found to be 0%; at 24 °C it was 7%, at 27 °C germination it was 14%, at 30 °C it reached 15%, decreasing thereafter to 3% and 1% at 35 °C and 40 °C, respectively [31], which might highly increase the recruitment of new plants. Incidentally, in the bioassay reported by Milberg et al. the temperature of incubation never exceeded 18.5 ± 2.0 °C.

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
