# Peer review of "Responses of Germination to Light and to Far-Red Radiation—Can they be Predicted from Diaspores Size?"

_data, 2020_

Round 1

Reviewer 1 Report

The objective of this paper was to study if diaspores size, departure from sphericity and type (fruit or seed) can be used to predict germination under light or dark conditions or under far-red enhanced radiation.

The paper is interesting, very complete and deserve publication in Data. In my opinion, some improvement could render the final result even more attractive for the reader.

The abstract should be rewritten. This section do not objectively reflect the most significant findings.

Please, write the scientific names in cursive throughout the manuscript.

-It lacks criticism regarding the experimental results, while this focus would be essential. Please, discuss the results with previous works related with this work in the Discussion section.

Author Response

As pdf file.

Reviewer 2 Report

 This is an interesting paper devoted to large data analysis of light/far red light effect on seeds germination from large populations data. Authors trying to distinguish between mass, size and volume, what is definitely a very interesting approach.

Finally, authors conclusions were supported by light scale analysis in many plant species based on Gorsky et al with combination with recent populations data.  

As major point, it is important to distinguish between seeds germination (dormancy broken) and seedlings establishment. What is not the same. Please, add some words  about this point.  

Minor points:

Line 14: “might be far-red inhibited” – do you mean far-red light?

I would suggest to also take in to account this paper in discussion part: “Türke M, Weisser WW (2013) Species, Diaspore Volume and Body Mass Matter in Gastropod Seed Feeding Behavior. PLoS ONE 8(7): e68788. doi:10.1371/journal.pone.0068788”.

Citation 2 contain basic paper from 2000 (20 years ago). It is better to find more recent one because ether are so many update available, for example:

Baranoski, G. V., Kimmel, B. W., Varsa, P., & Iwanchyshyn, M. (2019, October). Porosity effects on red to far-red ratios of light transmitted in natural sands: implications for photoblastic seed germination. In Remote Sensing for Agriculture, Ecosystems, and Hydrology XXI (Vol. 11149, p. 111490O). International Society for Optics and Photonics.

Lines 136-147: authors may add some explanation why small seeds require light for germination/seedlings establishment. Small seeds may have less storage compounds (carbohydrates) what is require for initial stage of seedlings establishment. That’s why they may need light to induce carbogydate production, what is require for early stage of seedlings growth.

Line 181: it is better to write more precisely, not far-red effect, but far-red light effect on germination.

Author Response

As pdf file.
